# Meta-Analysis of Survival and Development of a Prognostic Nomogram for Malignant Pleural Mesothelioma Treated with Systemic Chemotherapy

**DOI:** 10.3390/cancers13092186

**Published:** 2021-05-02

**Authors:** Rupesh Kotecha, Raees Tonse, Muni Rubens, Haley Appel, Federico Albrecht, Paul Kaywin, Evan W. Alley, Martin C. Tom, Minesh P. Mehta

**Affiliations:** 1Department of Radiation Oncology, Miami Cancer Institute, Baptist Health South Florida, Miami, FL 33176, USA; Mohammed.tonse@baptisthealth.net (R.T.); MuniR@baptisthealth.net (M.R.); HaleyA@baptisthealth.net (H.A.); MartinTo@baptisthealth.net (M.C.T.); MineshM@baptisthealth.net (M.P.M.); 2Herbert Wertheim College of Medicine, Florida International University, Miami, FL 33199, USA; 3Department of Medical Oncology, Miami Cancer Institute, Baptist Health South Florida, Miami, FL 33176, USA; FedericoA@baptisthealth.net (F.A.); PaulKa@baptisthealth.net (P.K.); 4Department of Medical Oncology, Cleveland Clinic Florida, Weston, FL 33331, USA; alleye@ccf.org

**Keywords:** mesothelioma, first line, meta-analysis, systematic review

## Abstract

**Simple Summary:**

Malignant pleural mesothelioma (MPM) is a rare cancer with an aggressive disease course. For patients who are medically inoperable or surgically unresectable, multi-agent systemic therapy remains an accepted standard-of-care around the world. Given the rare incidence of MPM and the disease’s aggressive nature, novel clinical trial designs are required. The purpose of this meta-analysis is to provide baseline summative survival estimates as well as evaluate the influence of prognostic variables to provide comparative estimates for future trial designs. In this study, a nomogram model was created to estimate survival with treatment with platinum-pemetrexed using covariates known to be associated with survival, including median age, gender, ECOG performance status, tumor stage, and tumor pathology subtype. Collaborative efforts can drive the change in the right direction, and appreciable progress has to be facilitated and newer trial designs may need to pave the way for future innovations in this rare disease.

**Abstract:**

(1) Purpose: Malignant pleural mesothelioma (MPM) is a rare cancer with an aggressive course. For patients who are medically inoperable or surgically unresectable, multi-agent systemic chemotherapy remains an accepted standard-of-care. The purpose of this meta-analysis is to provide baseline summative survival estimates as well as evaluate the influence of prognostic variables to provide comparative estimates for future trial designs. (2) Methods: Using PRISMA guidelines, a systematic review and meta-analysis was performed of MPM studies published from 2002–2019 obtained from the Medline database evaluating systemic therapy combinations for locally advanced or metastatic disease. Weighted random effects models were used to calculate survival estimates. The influence of proportions of known prognostic factors on overall survival (OS) were evaluated in the creation of a prognostic nomogram to estimate survival. The performance of this model was evaluated against data generated from one positive phase II study and two positive randomized trials. (3) Results: Twenty-four phase II studies and five phase III trials met the eligibility criteria; 2534 patients were treated on the included clinical studies. Ten trials included a platinum-pemetrexed-based treatment regimen, resulting in a pooled estimate of progression-free survival (PFS) of 6.7 months (95% CI: 6.2–7.2 months) and OS of 14.2 months (95% CI: 12.7–15.9 months). Fifteen experimental chemotherapy regimens have been tested in phase II or III studies, with a pooled median survival estimate of 13.5 months (95% CI: 12.6–14.6 months). Meta-regression analysis was used to estimate OS with platinum-pemetrexed using a variety of features, such as pathology (biphasic vs. epithelioid), disease extent (locally advanced vs. metastatic), ECOG performance status, age, and gender. The nomogram-predicted estimates and corresponding 95% CIs performed well when applied to recent randomized studies. (4) Conclusions: Given the rarity of MPM and the aggressive nature of the disease, innovative clinical trial designs with significantly greater randomization to experimental regimens can be performed using robust survival estimates from prior studies. This study provides baseline comparative values and also allows for accounting for differing proportions of known prognostic variables.

## 1. Introduction

Malignant pleural mesothelioma (MPM) is a rare cancer with an aggressive disease course associated with poor prognosis [1]. Its incidence in the United States is approximately 3000 new cases diagnosed annually, but is still increasing in the rest of the world, particularly Asia and Europe [2]. Due to its insidious presentation, most patients are diagnosed with locally advanced or metastatic disease-unamenable to radical resection. For patients who are medically inoperable or surgically unresectable, multi-agent systemic chemotherapy remains a current standard-of-care with a median survival of approximately 12 months [3].

Although recent data from the CheckMate 743 trial have demonstrated improved outcomes with first-line immunotherapy [4], the combination of cisplatin and pemetrexed is commonly utilized in the front-line setting worldwide [5,6]. Carboplatin has similar efficacy to cisplatin, with a favorable toxicity profile and ease of administration; therefore, it has often been used in combination with pemetrexed for a large proportion of MPM patients, especially the elderly [7]. The purpose of this meta-analysis is to provide baseline summative survival estimates as well as evaluate the influence of basic prognostic variables to provide comparative estimates for future trial designs.

## 2. Methods

### 2.1. Selection of Articles

The Preferred Reporting Items for Systematic reviews and Meta-Analyses (PRISMA) criteria were followed in conducting this systematic review and meta-analysis [8]. The article selection was performed by searching the MEDLINE (PubMed) and Cochrane electronic bibliographic databases for first-line systemic therapy combinations for patients with locally advanced or metastatic MPM. To ensure a comprehensive initial search strategy, generic key words were used in the initial article screen: “mesothelioma” and “locally advanced” and “metastatic” and “first-line” and “systemic therapies” and “platinum/pemetrexed” and “experimental therapies”. Full text articles published in the English language were considered and no publishing date restrictions were used through February 2021.

The initial query identified 447 reports that were subsequently screened by thorough review of the article titles and abstracts, as necessary. Inclusion criteria were publication in the English language, phase II and phase III clinical trials with 10 or more patients evaluable and with published outcomes on the efficacy endpoints of interest. Publications that were available in abstract only form and those in languages other than English were excluded. Case reports and limited case series, preclinical trials, studies using locoregional interventions alone, and studies using second-line therapies, were all excluded. A manual review of the references of the articles that were retrieved was performed to identify additional relevant publications. The search strategy used for this meta-analysis and the methodology for study inclusion is illustrated in Appendix A.

The studies were divided by treatment regimen: platinum-pemetrexed-based treatment and other experimental therapies. The demographic data abstracted for this analysis included year of publication, acronyms of the study or study title, duration of the study period, type of study (phase II/III), primary and secondary endpoints, number of patients included, median age, sex (male/female), ECOG Performance status (0,1,2), tumor stage (loco-regional disease; stage I–III and metastatic disease; stage IV), and tumor pathology (epithelioid, biphasic, sarcomatoid). Overall survival (OS), 1-year and 2-year OS rates, progression free survival (PFS), and objective response rate (ORR) were the outcomes evaluated. The radiological response data included patients having complete response (CR), partial response (PR), stable disease (SD), progressive disease (PD), and disease control rate (DCR). The toxicity summary included patients with grade 3–4 toxicities and was subdivided into toxicity category (i.e., general, blood and lymphatic system, cardiac, gastro-intestinal, infections, respiratory, and skin).

### 2.2. Outcome Measures and Statistical Analysis

The primary outcomes were OS and PFS; extracted medians of these variables were transferred into a logarithm scale [9]. The random-effects model described by DerSimonian and Laird [10] was used for this analysis. For primary and secondary outcomes, corresponding forest plots were created. Study heterogeneity was assessed using I^2^ statistics. Values of 0–30%, 31–60%, 61–75%, and 76–100% indicated low, moderate, substantial, or considerable heterogeneity, respectively [11]. All analyses were performed in R (Version 4.0, R Foundation for Statistical Computing, Vienna, Austria). For identifying publication bias, funnel plots and the Egger test were used. Statistical significance of *p* < 0.05 indicated the presence of bias. To investigate the potential effects of each of the prognostic variables on OS, patient characteristics were also extracted from each study and included as predictors in the meta-regression model. Considered variables include median age, gender, ECOG performance status, tumor stage, and tumor pathology. The extent to which the meta-regression model explained heterogeneity of the effect among studies was quantified by the percentage reduction of between-study variability. Plot of residuals was used to check the adequacy of the meta-regression model. Nomograms were used to represent results of the meta-regression model, estimating survival time using the covariates. In developing the nomogram, we used model coefficients to assign points to characteristics and predictions from the model to map cumulative point totals. Finally, the nomogram was used to predict the overall survival outcomes reported in the positive phase II reports (STELLAR [12] study) and phase III studies (MAPS [13] and CheckMate 743 [4]) and compared to the original results to assess the model performance.

## 3. Results

Twenty-four phase II studies and five phase III trials were included in this meta-analysis with outcomes data collected on 2534 patients (Appendix A). Key patient characteristics, demographics, and treatment information were not uniformly or consistently reported across the literature. However, there was no publication bias detected (*p* > 0.05) across the included studies regarding the primary outcomes evaluated in this meta-analysis (Appendix A).

### 3.1. Demographic Data of Platinum-Pemetrexed Regimen

Ten trials (*n* = 1303 patients) included a platinum-pemetrexed-based treatment regimen with a median of 89 patients in each study (range: 11–302 patients) (Table 1). Across all studies, 81% were male, and the median age was 66 years (range: 59–72 years). The majority of patients (60%) had an ECOG status of 1. The patients diagnosed with loco-regional disease and metastatic disease were 35% and 47%, respectively. The majority of patients across all studies were epithelioid (80%), followed by biphasic (11%), and sarcomatoid (8%).

### 3.2. Treatment Outcomes, Radiological Response, and Toxicity Summary Data of Platinum-Pemetrexed Regimen

Treatment with a platinum-pemetrexed-based regimen resulted in a pooled PFS of 6.7 months (95% CI: 6.2–7.2 months) and an OS of 14.2 months (95% CI: 12.7–15.9) (Figure 1A,B).

Across all studies, the proportion of ORR was 24% (95% CI: 12–35%) and DCR was 73% (95% CI: 56–90%) (Table 2). Across all patients, the proportion of individual response rates for CR was 1.5% (95% CI: 1–4%), 19% PR (95% CI: 10–27%), 53% SD (95% CI: 37–69%), and 31% PD (95% CI: 14–48%).

The pooled estimates of the treatment-related toxicity outcomes for patients who received a platinum-pemetrexed regimen (Figure 2A-I) with grade 3–4 blood and lymphatic system toxicities were anemia 10% (95% CI: 8–13%), neutropenia 22% (95% CI: 15–30%), and thrombocytopenia 7% (95% CI: 5–10%). Cardiac toxicity was seen in 1% (95% CI: 0–3%), gastro-intestinal toxicity in 3% (95% CI: 1–5%), fatigue in 6% (95% CI: 3–12%), infections in 5% (95% CI: 3–6%), skin toxicity in 1% (95% CI: 0–3%), and nausea and vomiting in 6% (95% CI: 3–10%).

### 3.3. Demographic Data of Experimental Regimens

Nineteen trials tested 15 experimental chemotherapy regimens (*n* = 1231 patients) in negative phase II or III studies, with a median of 52 patients (range: 20–229 patients) in each study (Table A1). Across these studies, 75% were male, and the median age was 63 years (range: 55–72 years). Patients had an ECOG status of 0 (30%), 1 (60%), and 2 (10%). The patients diagnosed with loco-regional disease and metastatic disease were 39% and 34%, respectively. In these studies, the majority of patients had epithelioid subtype (76%), followed by biphasic (15%), and sarcomatoid (9%). 

### 3.4. Outcomes, Radiological Response, and Toxicity Summary Data of Experimental Regimens

Treatment with these experimental regimens resulted in a pooled estimate of PFS of 6.6 months (95% CI: 6.2–7.0 months) and OS of 13.5 months (95% CI: 12.6–14.6 months) (Figure 1C,D). Across all studies, the proportion of ORR was 31% (95% CI: 26–36%) and the DCR was 76% (95% CI: 69–84%) (Table A2). Responses using these experimental therapies were low: overall proportions for CR were 0.7% (95% CI: 0.3–1.6%), 29% PR (95% CI: 24–34%), 48% SD (95% CI: 42–55%), 22% PD (95% CI: 13–29%). 

The pooled toxicity estimates for patients who received experimental chemotherapy regimens (Figure 3A-I) resulted in blood and lymphatic system grade 3–4 toxicities, with anemia in 4% (95% CI: 2–7%), neutropenia in 21% (95% CI: 12–33%), and thrombocytopenia in 12% (95% CI: 6–24%). Cardiac toxicity was seen in 4% (95% CI: 2–9%), gastro-intestinal toxicity in 4% (95% CI: 2–7%), fatigue in 12% (95% CI: 10–15%), infections in 5% (95% CI: 3–7%), skin toxicity in 1% (95% CI: 0–3%), and nausea and vomiting in 9% (95% CI: 6–15%).

### 3.5. Development of a Prognostic Nomogram to Estimate Survival

Meta-regression analysis was used to estimate survival with treatment with platinum-pemetrexed using covariates known to be associated with OS, including median age, gender, ECOG performance status, tumor stage, and tumor pathology subtype (Figure 4). 

Unlike the aforementioned experimental regimens, two randomized phase III trials and one single-arm phase II trial have demonstrated promising outcomes in this disease entity. The Mesothelioma Avastin Plus Pemetrexed-cisplatin Study (MAPS) [13] evaluated cisplatin/pemetrexed/bevacizumab compared to cisplatin/pemetrexed, the STELLAR trial [12] evaluated the use of tumor-treating fields (TTFields) in addition to cisplatin/pemetrexed, and recently CheckMate 743 [4] evaluated nivolumab plus ipilimumab compared to cisplatin/carboplatin and pemetrexed. To evaluate the prognostic nomogram developed in this study, we compared the estimated outcomes using the patient populations enrolled onto these studies and the proportion of each of the covariates and compared the nomogram estimates with the published results. For the MAPS study, given the patient population in the experimental arm of the phase III study, the OS estimate from the nomogram was 15.76 months (95% CI: 13.96–17.81 months) compared to the reported 18.8 months in the study. Similarly, the OS estimate from the nomogram using the CheckMate 743 trial was 13.65 months (95% CI: 11.41–16.33 months) compared to 18.1 months reported in the experimental arm. Therefore, the results of the experimental arms of these two studies were outside the confidence interval estimate based on historical data and consistent with a positive outcome. For the STELLAR trial, the OS estimate from the nomogram was 16.95 months (95% CI: 10.49–27.38 months) and given the wide confidence interval, potentially could overlap with the 18.2 months reported in the study. 

## 4. Discussion

Since 2003, chemotherapy with cisplatin/carboplatin and pemetrexed has been a standard first-line therapy for the majority of newly diagnosed patients who have locally advanced and metastatic MPM [6]. Over the past 15 years, multiple studies have established the outcomes for MPM patients treated with this regimen including single-arm phase II trials [7,14,15], the experimental arms of randomized trials compared to cisplatin alone [6], and the control arms of randomized trials testing novel experimental regimens [4,13,17,18,19,20]. In total, 1303 patients have been treated with this regimen across 10 studies, the data of which were abstracted in this systematic review and meta-analysis to determine pooled estimates of a PFS of 6.7 months and an OS of 14.2 months. In fact, a similar number—1231 patients—have been treated with experimental regimens who showed no improved outcomes compared to these historical estimates, underscoring the need for novel therapeutic development in this space. Moreover, despite advances in this field with the addition of bevacizumab and immunotherapy, doublet chemotherapy remains to be commonly used in most parts of the world where mesothelioma incidence continues to rise. Although the addition of bevacizumab to first-line chemotherapy has been added to the national guidelines [13], this regimen has not received FDA approval. Moreover, in CheckMate 743, nivolumab and ipilimumab were compared to pemetrexed-platinum, and although the OS was extended in the experimental arm, subgroup analysis yielded important caveats [4]. For example, for patients with epithelioid histologies (75% of those enrolled), the 12-month OS rates were not as striking (66% vs. 69%). Similarly, for those patients with a PDL-1 < 1%, the Kaplan-Meier curves crossed with longer follow-up, yielding an overall hazard ratio of 0.94. Hence, the role of first-line chemotherapy continues to be evaluated in ongoing trials.

Randomized controlled trials (RCTs) are deemed the gold standard of clinical research [21]. Randomization is often recommended for endpoints with a higher risk of confounding and selection bias, and it has been shown to improve the ability of phase II results to accurately predict phase III success [22,23]. However, modifications to traditional randomized trial designs have been performed to improve their performance in clinical practice. For example, the permuted block randomization has been widely used; however, in this design, there exists a compromise between effective imbalance control with a small block size and accurate allocation target with large block size [24]. Several alternative randomization designs have been proposed, such as the maximal procedure, brick tunnel randomization, and block urn designs [25,26,27]. However, for cancers such as mesothelioma, there are several logistical constraints for patients with rare diseases, as well as accrual/drop-out issues for those randomized to standard arms with known historically poor outcomes. Therefore, in other similar rare disease entities with robust historical survival estimates, there has been a resurgence in the consideration of alternative clinical trial designs [28]. Bayesian randomized designs and multi-arm multi-stage designs are two different approaches for improving reliability by using patient outcomes [29]. The Bayesian design allocates a greater proportion of prospective patients to well-performing treatments, whereas the multi-arm multi-stage designs use pre-specified stopping boundaries to discontinue novel treatments due to lack of efficacy. Although the Bayesian randomized designs have been shown to be more effective than traditional RCTs in multi-arm studies, their efficiency improvements in two-arm studies have been modest, especially if the rate of accrual outpaces the event rate, since the latter is required to modified the “prior” in a Bayesian concept [29]. Some studies examined the effects of phase II designs for binary endpoints on subsequent phase III trials, and found that randomization is useful when interstudy variability is high or there is a tendency to underestimate the control response [30]. Therefore, there is continued need to develop novel methods of clinical study and pooling historical data may help in future with future trial designs. 

Meta-regression, the technique used in this study to develop the nomogram, is often used to assess the relationship between one or more covariates and a dependent variable. Similar approaches can be performed with a meta-analysis alone; however the covariates are at the level of the study rather than the level of the subject [31]. The differences that we need to address as we transition from using primary study data to meta-analysis for regression are similar to those for subgroup analyses. For example, in this meta-analysis, using meta-regression, we identified variables that were associated with OS and developed a nomogram to determine the influence of each of these on survival, including median age, gender, ECOG performance status, tumor stage, and tumor pathology. Using the nomogram, the overall survival was predicted as reported in the positive phase II and III studies and compared to the original result reported in these studies. 

In the MAPS study [13], the patient population in the experimental arm of the phase III study showed an OS estimate from the nomogram to be 15.76 months (95% CI: 13.96–17.81 months), as compared to the 18.8 months that was reported in the original study. Similarly, for another phase III study (CheckMate 743) [4], the OS estimate from the nomogram was 13.65 months (95% CI: 11.41–16.33 months), as compared to 18.1 months reported in the study. Based on the nomogram model developed from historical estimates, the OS reported for the positive phase III trials are outside of the 95% confidence interval range of the historical estimates; however, the predicted OS from the nomogram was also similar to the OS from the control arms in the original studies, indicating good performance. Interestingly, in the single-arm phase II STELLAR study [12], the OS estimate from the nomogram was 16.95 months (95% CI: 10.49–27.38 months), compared to 18.2 months. The OS reported in this study falls within the range of the 95% confidence interval predicted from the nomogram. This demonstrates the importance of patient numbers in phase II trials, as the effectiveness of a phase II trial cannot be measured due to the wide confidence interval, prompting well-powered confirmatory studies. A well-designed phase II trial with complete reporting of the trial design, patient eligibility, study endpoints, and statistical analyses may be reliable and applicable in rare diseases, such as MPM [32].

There are important limitations to our analysis that should be noted. Formally, any categorical variable should have specific outcome-specific data to optimize the performance of the meta-regression. For example, for gender, male-specific OS and female-specific OS should be calculated. Unfortunately, this was difficult to extract from existing publications, since this level of detail is seldom reported. Similarly, Brims et al. [33] developed a prediction model for MPM using variables like Hb, weight loss, and albumin, which was unable to be extracted from existing publications for this study but would likely improve the performance of the survival estimates. However, in this study, we included percentages as continuous variables in the meta-regression. Furthermore, individual patient-level data can also be used to enhance any created model and should be pursued in subsequent studies. Given this promising approach with study-level data, further projects using individual patient-level data should be performed.

## 5. Conclusions

Given the rare incidence of MPM and the aggressive nature of the disease course, innovative clinical trial designs with significantly weighted randomization to experimental regimens can be utilized using robust survival estimates from prior studies. This study provides baseline comparative values and also allows for accounting for differing proportions of known prognostic variables. Collaborative efforts can drive change in the right direction, and appreciable progress has to be facilitated. Newer trial designs may be needed to pave the way for future innovations in this rare disease.

## Figures and Tables

**Figure 1 cancers-13-02186-f001:**
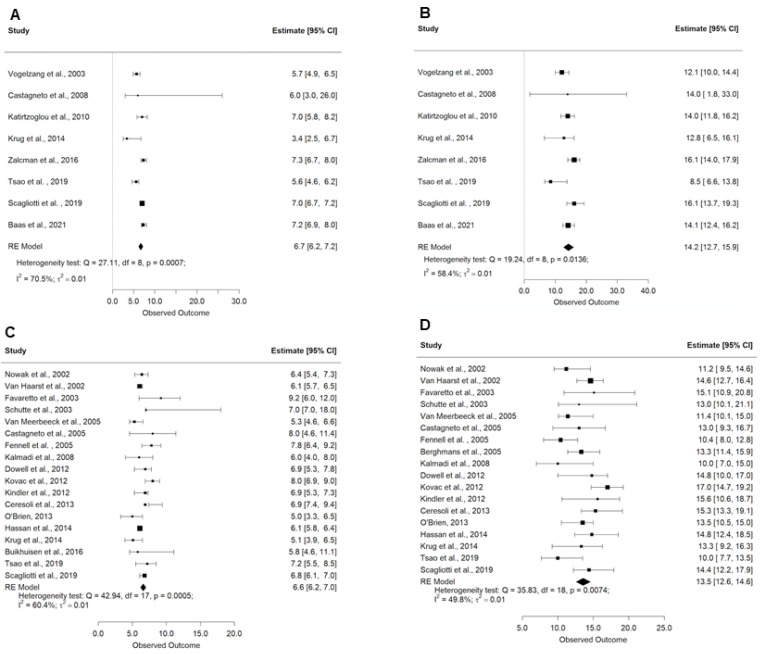
Forest plots demonstrating the (**A**) progression-free survival with platinum/pemetrexed; (**B**) overall survival with platinum/pemetrexed; (**C**) progression-free survival with other experimental therapies; and (**D**) overall survival with other experimental therapies. Squares indicate the proportions from individual studies and horizontal lines indicate the 95% confidence interval. The size of the data marker corresponds to the relative weight assigned in the pooled analysis using the random effects model. The diamond indicates the pooled proportion with 95% CI.

**Figure 2 cancers-13-02186-f002:**
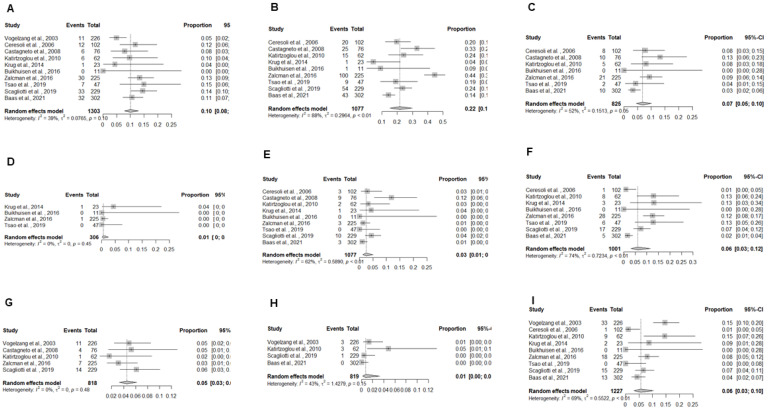
Forest plots demonstrating the toxicity outcomes for platinum/pemetrexed regimen based on toxicity category: (**A**) Anemia; (**B**) Neutropenia; (**C**) Thrombocytopenia; (**D**) Cardiac Toxicity; (**E**) Gastro-intestinal toxicity; (**F**) Fatigue; (**G**) Infections; (**H**) Skin toxicity; and (**I**) Nausea and vomiting. Squares indicate the proportions from individual studies and horizontal lines indicate the 95% confidence interval. The size of the data marker corresponds to the relative weight assigned in the pooled analysis using the random effects model. The diamond indicates the pooled proportion with 95% CI.

**Figure 3 cancers-13-02186-f003:**
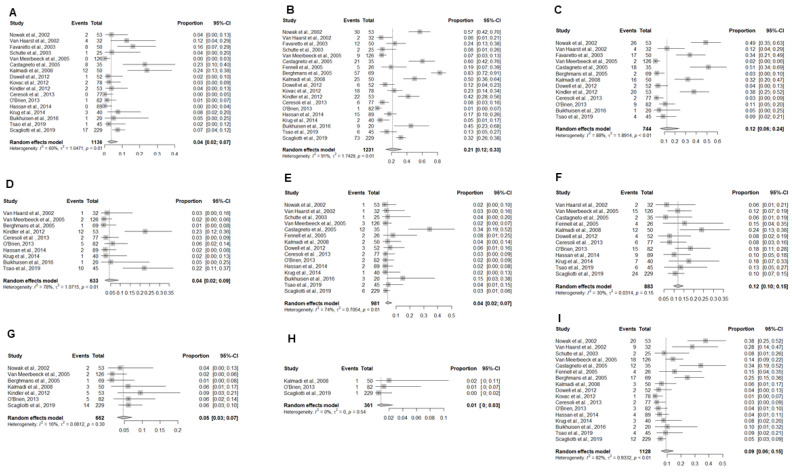
Forest plots demonstrating the toxicity outcomes for various experimental regimens: (**A**) Anemia; (**B**) Neutropenia; (**C**) Thrombocytopenia; (**D**) Cardiac Toxicity; (**E**) Gastro-intestinal toxicity; (**F**) Fatigue; (**G**) Infections; (**H**) Skin toxicity; and (**I**) Nausea and vomiting. Squares indicate the proportions from individual studies and horizontal lines indicate the 95% confidence interval. The size of the data marker corresponds to the relative weight assigned in the pooled analysis using the random effects model. The diamond indicates the pooled proportion with 95% CI.

**Figure 4 cancers-13-02186-f004:**
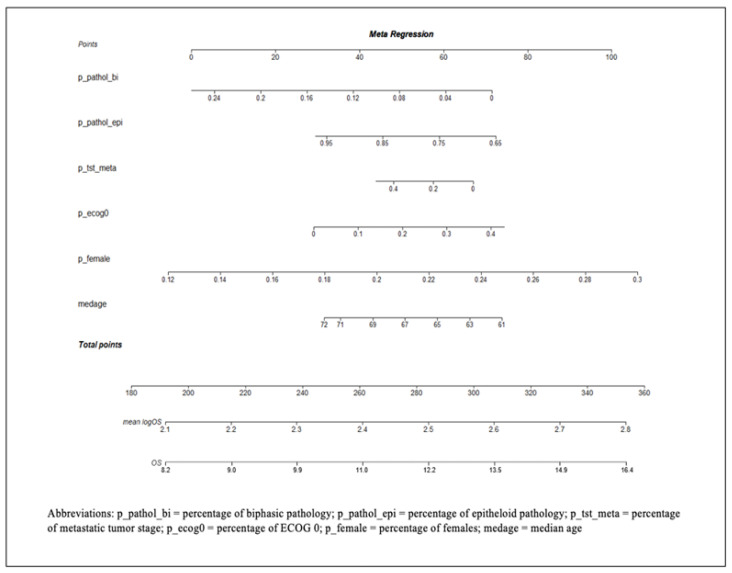
Nomogram model developed to predict overall survival (OS) in patients with malignant pleural mesothelioma treated with platinum/pemetrexed therapy. The mean log OS can be calculated by drawing a vertical line connecting the value of each variable with the point score at the top of the nomogram. The point scores for individual variables are then summed to get a total point score. This is then plotted along the total points line at the bottom of the nomogram. This line is projected to the mean log OS of the trial. Then the exponential of mean log OS is calculated to obtain the OS in months.

**Table 1 cancers-13-02186-t001:** Demographic data of malignant pleural mesothelioma patients treated with cisplatin/carboplatin and pemetrexed.

Author	Year	Acronymous	Duration	Type of Study	Treatment	Primary End Point	Secondary Endpoint	N	Median Age	Sex [No.(%)]		ECOG Performance Status [No.(%)]			Tumour Stage		Tumour Pathology		
										Male	Female	0	1	2	Loco-Regional (Stage I–III)	Metastatic(Stage IV)	Epithelioid	Biphasic	Sarcomatoid
Vogelzang et al. [6]	2003	NA	1999–2001	III	Cisplatin + Pemetrexed	OS	PFS, RR, duration of response	226	61	184 (81%)	42 (19%)	NA	NA	NA	73 (32%)	102 (45%)	154 (68%)	37 (16%)	18 (8%)
Ceresoli et al. [7]	2006	NA	2002–2005	II	Carboplatin + Pemetrexed	ORR	toxicity, TTP, and OS	102	65	76 (75%)	26 (25%)	33 (32%)	61 (60%)	8 (8%)	34 (33%)	49 (48%)	80 (78%)	8 (8%)	7 (7%)
Castagneto et al. [14]	2008	NA	2003–2005	II	Carboplatin + Pemetrexed	RR	OS, TTP, Toxicity	76	65	54 (71%)	22 (29%)	NA	NA	NA	27 (36%)	36 (48%)	57 (75%)	13 (17%)	3 (4%)
Katirtzoglou et al. [15]	2010	NA	2004–2007	II	Carboplatin + Pemetrexed	RR	OS, TTP	62	66	53 (86%)	9 (14%)	25 (40%)	37 (60%)	0	23 (37%)	17 (27%)	47 (76%)	NA	15 (24%)
Krug et al. [16]	2014	NA	NA	II	Cisplatin + Pemetrexed	PFS	OS, DCR, and safety/toxicity	23	66	20 (87%)	3 (13%)	7 (30%)	16 (70%)	0	NA	NA	16 (70%)	2 (9%)	11 (18%)
Buikhuisen et al. [17]	2016	NA	2009–2012	II	Cisplatin + Pemetrexed	RR	OS, PFS	11	59	10 (89%)	1 (11%)	NA	NA	NA	NA	NA	10 (89%)	1 (11%)	0
Zalcman et al. [13]	2016	MAPS	NA	III	Cisplatin + Pemetrexed	OS	PFS, QoL and safety	225	66	170 (76%)	55 (25%)	NA	NA	NA	NA	NA	182 (81%)	NA	NA
Tsao et al. [18]	2019	SWOG S0905	2011–2016	II	Cisplatin + Pemetrexed	PFS	OS, DCR, and safety/toxicity	47	72	40 (85%)	7 (15%)	NA	NA	NA	NA	NA	35 (74%)	12 (26%)	NA
Scagliotti et al. [19]	2019	LUME-Meso	2016–2018	III	Cisplatin + Pemetrexed	PFS	OS, ORR, DCR, QoL	229	66	169 (74%)	60 (26%)	98 (43%)	131 (57%)	NA	90 (39%)	105 (46%)	223 (97%)	6 (3%)	NA
Baas et al. [4]	2021	CheckMate 743	2016–2018	III	Cisplatin/Carboplatin + Pemetrexed	OS	PFS, ORR, DCR	302	69	233 (77%)	69 (23%)	124 (42%)	173 (57%)	NA	106 (35%)	149 (49%)	227 (75%)	39 (13%)	36 (12%)

Abbreviations: OS = overall survival; PFS = progression free survival; ORR = objective response rate; RR = response rate; TTP = time to progression; DCR = disease control rate; QoL = quality of life, ECOG = Eastern Cooperative Oncology Group; NA = not available.

**Table 2 cancers-13-02186-t002:** Treatment outcomes, radiological response, and toxicity summary for malignant pleural mesothelioma patients treated with cisplatin/carboplatin and pemetrexed.

Author	Year	N	OS (Months)	1 Yr. Survival Rates	2 Yr. Survival Rates	PFS (Months)	Objective Response Rate (ORR)	Radiological Response Rate [N (%)]					Toxicity Summary (Grade 3 and 4) N (%)									
								Complete Response	Partial Response	Stable Disease	Progressive Disease	Disease Control Rate (%)	Blood and Lymphatic System Disorders			Cardiac Disorders	Gastrointestinal Disorders	Fatigue	Infections	Respiratory Disorders	Skin Disorders	Nausea and Vomiting
													Anemia	Neutropenia	Thrombocytopenia							
Vogelzang et al. [6]	2003	226	12.1	6	NA	5.7	41.3	NA	NA	NA	NA	NA	11 (5%)	NA	NA	NA	NA	NA	11 (5%)	NA	3 (1%)	33 (15%)
Ceresoli et al. [7]	2006	102	12.7	6.5	NA	6.5	19	2 (2%)	17 (17%)	48 (47%)	33 (33%)	67	12 (12%)	20 (20%)	8 (8%)	NA	3 (3%)	1 (1%)	NA	NA	NA	1 (1%)
Castagneto et al. [14]	2008	76	14	NA	NA	6	25	3 (4%)	16 (21%)	29 (38%)	28 (37%)	63	6 (8%)	25 (33%)	10 (13%)	NA	9 (12%)	NA	4 (5%)	NA	NA	NA
Katirtzoglou et al. [15]	2010	62	14	NA	NA	7	29	0	18 (29%)	34 (56%)	10 (16%)	85	6 (10%)	15 (24%)	5 (8%)	NA	2 (3%)	8 (13%)	1 (1%)	NA	3 (5%)	9 (15%)
Krug et al. [16]	2014	23	12.8	NA	NA	3.4	10	0	2 (10%)	10 (50%)	8 (40%)	60	1 (4%)	1 (4%)	NA	1 (4%)	1 (4%)	3 (13%)	NA	NA	NA	2 (9%)
Buikhuisen et al. [17]	2016	11	18.5	NA	NA	8.3	18	NA	2 (18%)	8 (73%)	NA	91	0	1 (5%)	0	0	0	0	NA	0	NA	0
Zalcman et al. [13]	2016	225	16.1	NA	NA	7.3	NA	NA	NA	NA	NA	NA	30 (13%)	100 (44%)	21 (9%)	2 (1%)	3 (1%)	28 (13%)	7 (3%)	NA	NA	18 (8%)
Tsao et al. [18]	2019	47	8.5	NA	NA	5.6	20	NA	NA	NA	NA	NA	7 (15%)	9 (20%)	2 (4%)	0	0	6 (13%)	NAA	NA	NA	0
Scagliotti et al. [19]	2019	229	16.1	NA	NA	7	43	NA	98 (43%)	NA	NA	93	33 (14%)	54 (24%)	NA	NA	10 (4%)	17 (7%)	14 (6%)	NA	1 (1%)	15 (7%)
Baas et al. [4]	2021	302	14.1	8.1	3.8	7.2	43	0	129 (43%)	125 (41%)	14 (5%)	85	32 (11%)	43 (15%)	10 (3%)	NA	3 (2%)	5 (2%)	NA	NA	0	13 (4%)

Abbreviations: OS = overall survival; PFS = progression free survival; yr. = year; ORR = objective response rate; QoL = quality of life; NA = not available; N = number.

## Data Availability

No new data were created or analyzed in this study. Data sharing is not applicable to this article.

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
