# Peer review of "Meta-Analysis of Survival and Development of a Prognostic Nomogram for Malignant Pleural Mesothelioma Treated with Systemic Chemotherapy"

_cancers, 2021, doi:10.3390/cancers13092186_

Round 1
Reviewer 1 Report
Kotecha R et al. Meta-analysis of Survival and Development of a Prognostic 2 Nomogram for Malignant Pleural Mesothelioma Treated with 3 Systemic Chemotherapy
This manuscript is a meta-analysis of phase 2 and 3 clinical trials evaluating first-line systemic or experimental therapy in MPM. The authors aimed to provide baseline summative survival estimates as well as evaluate the influence of basic prognostic variables to provide comparative estimates for future trial designs. The main limitation to the nonagram proposed is the lack of patient-level data.
This study is well-written and designed. The tables and figures are well presented. As such I have only minor comments:
1) How is the staging determined? In the manuscript, it states locally advanced or metastatic?
2) Were other variables that influence prognosis considered? Brims et al developed a prediction model for prognosis in malignant pleural mesothelioma using decision tree analysis
Other variables used: Hb, weight loss. albumin
I suspect the variables above will be difficult to obtain from the papers included in the manuscript?
3) Could participant (n=) numbers be included into table 2.
Author Response
Response to Reviewer 1 Comments
We have carefully considered the reviewer’s comments and address our changes in the following point-by-point responses. Our responses and revisions are outlined in red font (which represents modifications in the revised manuscript). We appreciate the opportunity to utilize the expertise of our peer reviewers and hope this revision will make the article acceptable for publication in Cancers.
Reviewer 1 Comments: This manuscript is a meta-analysis of phase 2 and 3 clinical trials evaluating first-line systemic or experimental therapy in MPM. The authors aimed to provide baseline summative survival estimates as well as evaluate the influence of basic prognostic variables to provide comparative estimates for future trial designs. The main limitation to the nomogram proposed is the lack of patient-level data.
This study is well-written and designed. The tables and figures are well presented.
Response: We thank the reviewer for their encouraging comments. In the limitations paragraph, we have acknowledged that individual patient-level data can be used to enhanced any prognostic model. Unfortunately, this is difficult to obtain given the different study sponsors and the lack of a clear mechanism for obtaining each of these data sets (including international data) for pooled summation.
As such I have only minor comments:
Point 1) How is the staging determined? In the manuscript, it states locally advanced or metastatic?
Response 1: We thank the reviewer for their comments. The following criteria were used to assess staging in the manuscript: all stage I-III disease was classified as loco-regional, and all stage IV disease was classified as metastatic. We have included this clarification in the revised manuscript on Page 2 Methods section (highlighted in red).
Point 2) Were other variables that influence prognosis considered? Brims et al developed a prediction model for prognosis in malignant pleural mesothelioma using decision tree analysis
Other variables used: Hb, weight loss. albumin
I suspect the variables above will be difficult to obtain from the papers included in the manuscript?
Response 2: We appreciate the reviewers comments. We used variables such as median age, gender, ECOG performance status, tumor stage (loco-regional or metastatic), and tumor pathology subtype (Epithelioid, Biphasic & Sarcomatoid) for the nomogram model to predict overall survival. We did consider other variables such as Hb, weight loss, and albumin, however it was impossible to obtain from the manuscripts evaluated in this study with enough clarity for pooled analysis. We have included this in our limitation section on Page 9 (highlighted in red).
Point 3) Could participant (n=) numbers be included into table 2.
Response 3: We appreciate the reviewers comments. Participant numbers have been included in Table 2 on Page 5 (highlighted in red).

Reviewer 2 Report
This is a well-written manuscript including a valuable addition to methods assessing the prognosis of MPM. Two (relatively small) issues: 1. MPM is usually diagnosed when the disease has progressed loco-regionally (metastatic disease relatively rare and frequently noticed after first-line treatment. and 2. The authors repeatedly state that platinum/pemetrexed remains the current standard. The recent comparative data obtained with first-line with immunotherapy suggest that this statement is too strong.
Author Response
Response to Reviewer 2 Comments
We have carefully considered the reviewers’ comments and address our changes in the following point-by-point responses. Our responses and revisions are outlined in red font (represents modifications in the revised manuscript). We appreciate the opportunity to utilize the expertise of our peer reviewers and hope this revision will make the article acceptable for publication in Cancers.
This is a well-written manuscript including a valuable addition to methods assessing the prognosis of MPM.
We thank the reviewer for their encouraging comments.
Point 1. MPM is usually diagnosed when the disease has progressed loco-regionally (metastatic disease relatively rare and frequently noticed after first-line treatment.
Response 1: We appreciate the reviewer’s comments. We agree with the reviewer that a majority of patients have locally-advanced disease rather than distant metastases. Given that the treatment paradigm is the same for these groups of patients, they are enrolled onto the same clinical trials and therefore were included in this study. We have also clarified in the methods section the stages of the patients included, updated Table 1 to reflect this, as well as used this variable in the creation of the prognostic nomogram.
Point 2. The authors repeatedly state that platinum/pemetrexed remains the current standard. The recent comparative data obtained with first-line with immunotherapy suggest that this statement is too strong.
Response 2: We appreciate the reviewer’s comments. We agree that the statement platinum/pemetrexed remains the standard of care is too strong, especially after comparative data the first-line immunotherapy from the CheckMate trial. We have revised this statement in the various parts of the manuscript; in the introduction section on Page 2, we have removed this statement and in the discussion section on page 8, we have modified the statement which is highlighted in red.
